# MicroRNAs of Milk in Cells, Plasma, and Lipid Fractions of Human Milk, and Abzymes Catalyzing Their Hydrolysis

**DOI:** 10.3390/ijms232012070

**Published:** 2022-10-11

**Authors:** Ivan Yu. Kompaneets, Evgeny A. Ermakov, Valentina N. Buneva, Georgy A. Nevinsky

**Affiliations:** Institute of Chemical Biology and Fundamental Medicine of Siberian Branch of Russian Academy of Sciences, 8 Lavrentiev Ave., Novosibirsk 630090, Russia

**Keywords:** human milk, abzymes, microRNA hydrolysis

## Abstract

Human milk provides neonates with various components that ensure newborns’ growth, including protection from bacterial and viral infections. In neonates, the biological functions of many breast milk components can be very different compared with their functions in the body fluids of healthy adults. Catalytic antibodies (abzymes) that hydrolyze peptides, proteins, DNAs, RNAs, and oligosaccharides were detected, not only in the blood sera of autoimmune patients, but also in human milk. Non-coding microRNAs (18–25 nucleotides) are intra- and extracellular molecules of different human fluids. MiRNAs possess many different biological functions, including the regulation of several hundred genes. Five of them, miR-148a-3p, miR-200c-3p, miR-378a-3p, miR-146b-5p, and let-7f-5p, were previously found in milk in high concentrations. Here, we determined relative numbers of miRNA copies in 1 mg of analyzed cells, lipid fractions, and plasmas of human milk samples. The relative amount of microRNA decreases in the following order: cells ≈ lipid fraction > plasma. IgGs and sIgAs were isolated from milk plasma, and their activities in the hydrolysis of five microRNAs was compared. In general, sIgAs demonstrated higher miRNA-hydrolyzing activities than IgGs antibodies. The hydrolysis of five microRNAs by sIgAs and IgGs was site-specific. The relative activity of each microRNA hydrolysis was very dependent on the milk preparation. The correlation coefficients between the contents of five RNAs in milk plasma, and the relative activities of sIgAs compared to IgGs in hydrolyses, strongly depended on individual microRNA, and changed from −0.01 to 0.80. Thus, it was shown that milk contains specific antibodies (abzymes) that hydrolyze microRNAs specific for human milk.

## 1. Introduction

Short, non-coding microRNAs (or miRNAs; 18–25 nucleotides) are intra- and extracellular molecules that are present in various human and animal fluids [1,2]. MicroRNAs have many different biological functions, including the regulation of up to several hundred genes [3,4]. Different changes in microRNAs (microRNA-regulated gene networks) can result in the realignment in the expression of many genes in different cells. It was shown that human milk could contain a few dozen to thousands of various microRNAs [4,5,6,7,8,9,10,11,12,13,14,15,16,17,18,19]. MicroRNAs in a mother’s milk have important functions for lactating breasts and infants [5]. The known data strongly support the idea that milk microRNAs enter the systemic circulation of infants, and have tissue-specific developmental and immunoprotective functions [5]. According to the literature data, human milk contains about 1400 mature miRNAs [5,6,7,8,9]. Based on the estimation of the value of the number of real-time polymerase chain reaction (PCR) cycles at which fluorescence exceeds the threshold value, it was concluded that miR-148a-3p, miR-200c-3p, miR-378a-3p, miR-146b-5p, and let-7f-5p are contained in human milk in high concentrations [5,6,7,8,9]. It was interesting to quantify and compare the relative content of these five microRNAs.

Antibodies (Abs) against chemically stable analogues of transition states of reactions, and natural antibodies (abzymes) having catalytic activities, are well described in the literature (reviewed in [11,12,13,14,15,16,17]). The spontaneous and antigen-stimulated evolution of autoimmune (AI) diseases (ADs) results in the production of abzymes (ABZs) against polysaccharides, lipids, peptides, proteins, RNAs, and DNAs and their complexes [11,12,13,14,15,16,17]. In the blood of AD patients were found different abzymes that act directly against antigens mimicking the conformations of transition states of chemical reactions. Moreover, secondary anti-idiotypic Abs (abzymes) in the active sites of different enzymes were also detected; their existence may be explained on the basis of Jerne’s anti-idiotypic network model [18]. The appearance of ABZs in the blood clearly indicates the beginning of AI processes in mammalians [11,12,13,14,15,16,17]. Abzymes (IgGs, IgMs, and IgAs) that hydrolyze RNAs and DNAs [19,20,21,22,23], polysaccharides [24,25,26], oligopeptides, and proteins [27,28,29,30,31,32,33], have been found in the blood sera of patients with different ADs and some viral diseases [11,12,13,14,15,16,17].

As a result of the absence of apparent immunization, the existence of any ABZs in people without any immune diseases was not considered to be possible. For example, auto-abzymes were not detected in healthy people and patients who did not demonstrate very severe infringements on their immune status [11,12,13,14,15,16,17].

A particular group of healthy people are pregnant and lactating females. Women’s milk contains different Abs (IgGs, IgAs, sIgAs, and IgMs), of which sIgAs are the major component (> 85–90%) [34,35]. The origin of milk IgGs is still debated; they could be partially produced locally by specific mammary gland cells, or partially moved from the circulation system of female blood [34]. IgAs are synthesized by women’s mammary gland B-lymphocytes [35]. IgA antibodies are produced by plasma cells in the mammary stroma; then, they are assembled to dimeric sIgAs on the basolateral surface of the epithelium [36]. During lactation, B cells that are stimulated by antigens in Peyer’s patches switch from IgM production to dimeric IgA, and migrate to the mammary gland [37,38].

The immune system of neonates during the first 4–6 months of life is immature: new-borns’ mucous surfaces and respiratory and gastroenterological tracts are still poorly filled with antibodies [39]. Infants begin to produce antibodies in the intestine in the first 3–5 months of life. However, neonates are well protected by antibodies of their mother’s milk (passive immunity), which cover mucous membranes with Abs against bacterial, viral, and other components [39]. Breast milk sIgAs are present in high concentrations and are active for at least 7–8 months after birth; they play an important role in maintaining the passive humoral response [40].

Women during pregnancy, and after the beginning of lactation, very often demonstrate a sharp exacerbation of AI reactions that is similar to what is found in typical autoimmune pathologies, including anti-phospholipid syndrome, systemic lupus erythematosus (SLE), multiple sclerosis (MS), thyroiditis, renal insufficiency, etc. [41,42,43,44,45]. Pregnancy realizes a range of specific changes in the immune system, leading to an increased risk of exacerbations of several diseases, and adverse maternal and fetal outcomes, including preeclampsia, fetal loss, and preterm birth [46,47]. SLE is often a disease that occurs during pregnancy [48,49], which sometimes leads to harmful situations for the mother and fetus [50]. The incidence of SLE exacerbation usually occurs during pregnancy, and within three months after delivery [48,49,51].

There is sometimes a remission of multiple sclerosis in the third trimester of pregnancy, but the disease worsens in the first postpartum period [52]. Autoimmune thyroid reactions are found in approximately 18% of pregnant women [53]. It is important to emphasize that many different ADs may be “triggered” or “activated” in healthy females during their pregnancy, and especially after childbirth [25,41,42,43,44,45,54,55,56]. Pregnancy and the onset of lactation are special periods that are associated with changes in the immune system of women [12,13,14,15,16,17,54,55]. These changes lead to the synthesis of various autoantibodies and abzymes in the blood and milk of women. The existence of abzymes in the blood and milk indicates the presence of AI reactions in women.

It was proposed earlier that different ADs may originate from specific defects of hematopoietic stem cells [57]. It was later shown that the development of various ADs occurs as a result of specific changes in the differentiation profiles of bone marrow stem cells (HSCs) [58,59,60,61,62].

Spontaneous and accelerated antigen development by SLE in SLE-prone MRL-lpr/lpr mice [58,59,60], and multiple sclerosis in EAE-prone C57BL/6 mice [61,62], leads to very similar specific reorganizations of their immune systems, which is bound with a production of abzymes that hydrolyze DNA and proteins.

Very similar changes (as in mice with deep SLE) in the differentiation profiles of HSCs were revealed in lactating mice [58,59,60]. Such changes in lactating mice are usually temporary, and return to normal after 1–3 months. In contrast, there are further changes in the differentiation profiles of mice diseased with SLE during the deepening of the pathology [58,59,60]. As in mice with deep SLE pathology, the changes in the differentiation profiles in lactating mice lead to the production of abzymes with high catalytic activity.

As shown in a number of studies, the abzymes of AD patients have a large number of different hydrolytic enzymatic activities. Small subfractions of milk polyclonal IgGs and sIgAs also split RNAs, DNAs [54,55], all nucleotides (NMPs, NDPs, and NTPs) [56]; they also possess amylase [25] and phosphatase [56] activities. Human milk is a unique source of abzymes, with not only hydrolytic but synthetic functions, including phosphorylation of lipids [63,64], and the synthesis of more than 15 different milk proteins [65,66] and polysaccharides [67,68], which were not found in patients with ADs. Human sIgAs and IgGs possess significantly higher enzymatic activities compared to abzymes of AD patients [12,13,14,15,16,17,54,55,56,57,58,59,60,61,62,63,64,65,66,67,68].

Initially, it was shown that antibodies (abzymes) from the blood of patients with multiple sclerosis [69], systemic lupus erythematosus [70], and schizophrenia [71,72] effectively hydrolyze four microRNAs that are characteristic of these diseases: miR-9-5p, miR-219-2-3p, miR-137, and miR-219a-5p. As noted above, human milk contains many different microRNAs, including these four microRNAs. It has recently been shown that sIgAs and IgGs of human milk effectively hydrolyze four microRNAs that are characteristic of immune diseases: miR-9-5p, miR-219-2-3p, miR-137, and miR-219a-5p [73,74]. However, these four microRNAs are more typical for patients with various ADs. Five microRNAs are most characteristic in human milk: miR-148a-3p, miR-200c-3p, miR-378a-3p, miR-146b-5p, and let-7f-5p [5,6,7,8,9,10].

It was interesting to understand whether there are abzymes in human milk that hydrolyze specific milk miRNAs, and whether they are sequence specific. Taking this into account, this study carried out for the first time a quantitative analysis of these microRNAs, and an assessment of the relative activities of human milk sIgAs and IgGs in the hydrolyses of these five microRNAs.

## 2. Results

### 2.1. Isolation and Quantification of MicroRNAs

Five miRNAs were isolated and quantitatively analyzed in seven preparations of human milk lipid fractions, cells, and plasma (number of copies in 1 mg of the analyzed sample (NC/mg)). Data on the content of five different microRNAs in individual preparations of plasma, cells, and lipid fractions are given in Table 1.

The average content of different microRNAs per 1 mg of three samples (milk lipid fraction, cells, and plasma) decreased in the following order (NC/mg): let-7f-5p (2.6 × 10^10^) ≈ miR-146b-5p (2.4 × 10^10^) > miR-200c-3p (2.6 × 10^9^) > miR-148a-3p (2.1 × 10^8^) > miR-378a-3p (8.7 × 10^7^) (Table 2).

The relative average content of all five microRNAs averaged over three types of different samples varied greatly (NC/mg): lipid fraction (1.36 × 10^10^), cells (1.25 × 10^10^), and plasma (0.18 × 10^10^) (Table 2). In addition, the average relative content of five different microRNAs in the three types of analyzed samples was very different. For example, the content of let-7f-5p decreased in the following order (NC/mg): cells (4.6 × 10^10^), lipid fraction (2.5 × 10^10^), and plasma (6.2 × 10^9^) (Table 2). At the same time, maximal content of miR-146b-5p was observed in lipid fraction (4.6 × 10^10^ NC/mg), then in cells (1.2 × 10^9^), and plasma (8.5 × 10^8^). The average content of miR-200c-3p in all three fractions was somewhat comparable (NC/mg): cells (4.0 × 10^9^), plasma (2.0 × 10^9^), and lipid fraction (1.8 × 10^9^) (Table 2). The maximum content in plasma was found for let-7f-5p (6.2 × 10^9^ NC/mg) and miR-200c-3p (2.0 × 10^9^ NC/mg), while the content of other microRNAs was much lower (Table 2). The ranges of change for all estimated parameters, their average values, as well as the values of the medians (M) and interquartile ranges (IQRs), are provided in Table 2.

An analysis of the correlation coefficients (CCs) was carried out for the content of five types of miRNAs (miR-148a-3p (parameter 1 (p1)), let-7f-5p (p2), miR-146b-5p (p3), miR-200c-3p (p4), and miR-378a-3p (p5)) in individual preparations of plasma, cells, and lipid fractions. Positive CCs between the five miRNAs in milk plasma varied from 0.14 to 0.92. However, three CCs were negative: p1–p3 (−0.005), p2–p4 (−0.19), and p2–p5 (−0.07) (Table 1). Positive correlation coefficients between the five microRNAs in milk cells varied from 0.04 to 0.99, and there were two negative values, p1–p5 (−0.17) and p2–p5 (−0.37) (Table 1). Nine positive CCs in the case of milk lipid fractions varied from 0.1 to 0.84, and the tenth coefficient was negative—p2–p5 (−0.32) (Table 1). Interestingly, in all three fractions (plasmas, cells, and lipid fractions), a negative correlation was observed between parameters p2 (let-7f-5p) and p5 (miR-378a-3p): −0.07–−0.37.

It was interesting how the content of each specific microRNA correlated in three different milk fractions. It turned out that the content of miR-148a-3p (0.49–0.94), let-7f-5p (0.18–0.3), miR-146b-5p (0.12–0.62), and miR-378a-3p (0.25–0.92) in plasma, cells and lipid fractions were characterized by positive CCs (Table 1). Three CCs that characterized the content of miR-200c-3p in three preparations turned out to be very different: 0.54 (cell–lipid fractions), 0.003 (plasma–lipid fractions), and −0.22 (plasma–cell fractions) (Table 1).

The statistical differences in the content of almost all microRNAs in the cell fractions were significant (*p* = 0.001–0.003), except for 148a-3p-278a-3p and 148a-3p-278a-3p (*p* = 0.79). A similar situation was observed in the lipid fractions (*p* = 0.001–0.04), except for the following two pairs of parameters: 148a-3p-278a-3p and 146b-5p-let-7f-5p (*p* = 0.1–0.96). The statistical significance of differences in the content of most microRNAs in milk plasma was also high (*p* = 0.001–0.01), except for 146b-5p-200c-3p and 148-3p-378-3p (*p* = 0.43–0.87). Thus, a quantitative analysis of the content of five microRNAs in plasma, cells, and lipid fractions of human milk was carried out. At the same time, no uniformity in the content of all five miRNAs in each of the three milk fractions (plasma, cells, and lipid fractions) was found. The content of each microRNA in each of the fractions turned out to be specific.

### 2.2. Purification and Characterizing of IgGs and sIgAs

It was shown that IgGs and sIgAs from the sera and milk of lactating mothers have several different catalytic activities ([12,13,14,15,16,17] and references therein). Methods of purification and characterization, including electrophoretic homogeneity of the IgG (150 kDa, H_2_L_2_: two light (L) and two heavy (H) chains) and sIgA (340 kDa, (H_2_L_2_)_2_SJ: four light and four heavy chains, secretory (S) and join (J) components) preparations used in this study, were described in [73,74]. In addition, the relative activities (RAs) of seven individual IgGs and sIgAs from the milk of healthy lactating mothers in hydrolyzing four miRNAs (miR-9-5p, miR-137, miR-219-2-3p, and miR-219a-5p) were characterized. Moreover, it was shown that the IgG and sIgA preparations used in this study do not contain impurities of canonical RNases [73,74]. In this study, we first analyzed the RAs of the same IgGs and sIgAs in the cleavage of five microRNAs: miR-148a-3p, miR-200c-3p, miR-378a-3p, miR-146b-5p, and let-7f-5p.

### 2.3. Hydrolysis of MicroRNAs

The relative activities in the hydrolyses of five microRNAs were analyzed using seven IgG and sIgA preparations that were isolated from milk plasma, as described in [73,74]. Typical patterns of the splitting of miR-146b-5p and miR-148a-3p by seven milk sIgAs and IgGs are shown in Figure 1. Three out of seven IgG preparations weakly hydrolyzed miR-146b-5p (Figure 1A). The major cleavage sites of miR-146b-5p in the case of IgGs, with numbers 3 and 4, are 10C-11U, 9C-10C, and 6A-7U, while other IgGs more weakly cleaved this miRNA in these sites (Figure 1A). Interestingly, four out of seven IgG preparations (especially with number 7) effectively hydrolyzed miR-146b-5p at the 18A-19A and 16U-17C sites. Exactly, these 18A-19A and 16U-17C sites of hydrolysis were found to be the most typical and common for all seven sIgA preparations (Figure 1A). The characteristic sites for IgGs in the case of sIgAs should be attributed to moderate hydrolysis sites.

There were many more major sites for the hydrolysis of miR-148a-3p by IgGs and sIgAs than for miR-146b-5p. For most IgGs and sIgAs, the following sites can be classified as major: 12A-13U, 11C-12A, 9G-10A, 6C-7A, 5U-6C, and 3U-4U (Figure 1B). At the same time, for some IgGs and sIgAs, individual major hydrolysis sites were observed, which should be classified as average or minor in the case of other antibody preparations.

This was pronounced especially in the case of sIgAs, with numbers 4 and 5 that very effectively hydrolyzed miR-148a-3p at the 14C-15A site (Figure 1B).

The only common major site of miR-200c-3p hydrolysis for IgGs and sIgAs antibodies was the 5A-6G site (Figure 2A). Four additional major sites were observed for three sIgA preparations, with numbers 1, 4, and 5: 11G-12G, 10U-11G, 9A-10U, and 6G-7U. For sIgA7, IgG1, IgG3, and IgG5, there was a pronounced average hydrolysis site—14C-15C. Interestingly, sIgA7 and IgG2–IgG6, in addition to efficient hydrolysis of miR-200c-3p at the 5A-6G site, split this microRNA at many other sites with approximately comparable efficiency (Figure 2A).

For all IgG preparations, the main major site of let-7f-5p hydrolysis was 7U-8G (Figure 2B). In addition, all IgG preparations effectively hydrolyzed this microRNA at two sites: 11A-12G and 10U-11A. Other sites of let-7f-5p hydrolysis in the case of some IgGs should be classified as moderate or minor. For seven sIgA preparations, no pronounced common sites of let-7f-5p hydrolysis were observed (Figure 2B). Three IgGs (with numbers 1, 4, and 5) very effectively cleaved this RNA at the 5U-6A site, while in the case of sIgAs, this site was a minor one. The 7U-8G site was major for four sIgA preparations with numbers 2, 3, 6, and 7 (Figure 2B).

Figure 3 demonstrates the patterns of the hydrolysis of miR-378a-3p by IgG and sIgA antibodies.

There were four moderate or minor sites found of miR-378a-3p splitting by IgGs and sIgAs: 20U-21C, 16C-17A, 15U-16C, and 11A-12G. At the same time, the 7A-8C, 6G-7A, 5A-6G, and 4A-5A sites were major in the case of two sIgAs, but they were average or minor for all other antibodies (Figure 3).

Table 3 shows the relative activities of seven individual IgG and sIgA preparations, and the average values for these preparations in the hydrolysis of five microRNAs. Several of the antibody sets did not meet the Gaussian normal distribution. Considering this, for all sets of parameters, we calculated the median (M) and interquartile ranges (IQR) (Table 3).

Interestingly, the hydrolyses of miR-148a-3p and miR-200c-3p by sIgAs, on average, were 1.7–1.9 times more active than by IgGs (Table 3).

The opposite situation was observed for miR-146b-5p, which was hydrolyzed by IgGs approximately 1.8 times more efficiently than by sIgAs. The average activities of sIgAs and IgGs in the hydrolyses of miR-378a-3p and let-7f-5p were comparable (Table 3). The most increased hydrolysis by sIgAs and IgGs, on average, was observed for let-7f-5p, and the lowest hydrolysis rate was found for miR-378a-3p (Table 3).

All correlation coefficients (CCs) for seven IgG preparations in the hydrolyses of five microRNAs were positive, and varied from 0.17 to 0.83 (Table 3). Some CCs for seven sIgA preparations in the hydrolyses of five microRNAs were positive, and varied from very low (0.04) to very high (0.98). sIgAs showed a weak negative correlation (−0.08) in the hydrolyses of miR-378a-3p and let-7f-5p. Interestingly, IgGs and sIgAs showed a negative correlation in the hydrolysis of miR-148a-3p (−0.07), miR-200c-3p (−0.41), and miR-378a-3p (−0.09). However, a high positive correlation (0.87) was observed in the hydrolysis by IgGs and sIgAs of two other microRNAs: let-7f-5p and miR-146b-5p (Table 3).

In the case of IgG preparations, a significant difference in the relative activities was observed in the hydrolysis of all five miRNAs (*p* = 0.002–0.03) (Table 3). The statistical difference of sIgA preparations in the hydrolyses of five microRNAs was also significant (*p* = 0.002–0.03), except for miR-148a-3p–miR-200c-3p, miR-148a-3p–miR-let-7f-5p (*p* = 0.70), and miR-200c-3p–miR-let-7f-5p (*p* = 0.41).

For IgG and sIgA preparations, a statistical difference was observed in the hydrolyses of three microRNAs: miR-148a-3p, miR-378a, and miR-146b (*p* = 0.002–0.03); however, in the case of two RNAs, this was not observed: miR-200c-3p and miR-let-7f-5p (*p* = 0.52–0.70).

All antibodies were isolated from milk plasma. Therefore, we compared CCs between concentrations of various microRNAs in plasma (Table 4).

All CCs were positive but very different, and varied from 0.007 for miR-200c-3p and let-7f-5p, to 0.99 in the case of miR-148a-3p and miR-378a-3p (Table 4). It could be expected that the relative activities of antibodies in the hydrolysis of each of the five RNAs would correlate with the relative concentrations of these microRNAs in milk plasma. However, this turned out to be far from the case. The CCs of the concentrations of miR-148a-3p (−0.01) and miR-200c-3p (−0.05) in plasma with the efficiency of their hydrolysis by IgGs turned out to be weakly negative. The rest of the CCs were positive: 0.41–0.79 (Table 5). In the case of sIgAs, all CCs were positive, and varied from 0.03 to 0.8.

### 2.4. Spatial Structures of Five miRNAs

The spatial structures of five miRNAs having minimal free energy were calculated. The relative amount (%) of every product of each microRNA hydrolysis by individual IgGs and sIgAs was calculated. Then, using the data of three independent experiments for each IgG and sIgA sample, the average percentage of every product that corresponded to seven milk plasma IgG and IgA preparations was calculated. Figure 4, Figure 5 and Figure 6 show the locations of hydrolysis sites in the spatial structures of five microRNAs in the case of IgG and sIgA antibodies. As indicated above and in Figure 1, Figure 2 and Figure 3, antibodies from milk plasma of various donors hydrolyzed five microRNAs with different efficiencies, and in some cases, at different sites. Taking this into account, Figure 4, Figure 5 and Figure 6 show averaged data on the efficiency of the hydrolysis of five microRNAs by seven IgG and sIgA preparations at each of the sites. Pronounced major hydrolysis sites of each microRNA, in the case of only some antibody preparations, are indicated in brackets.

The main sites for more efficient cleavage of miR-148a-3p by IgG and sIgA antibodies are located in the specific loop of this microRNA (Figure 4A,B). In the case of seven sIgAs, the hydrolyses at five sites of the loop were very different. Some sIgA preparations hydrolyzed this microRNA at these sites 1.3–3.0 times more efficiently in comparison with the average values for all seven sIgAs (indicated in brackets) (Figure 4A). The hydrolysis efficiencies of miR-148a-3p by the seven IgG preparations were more comparable (Figure 4B).

Four of the six hydrolysis sites of miR-200c-3p are also located in the loop of this RNA, but they can be classified as moderate cleavage sites in the case of IgGs (Figure 4D). Several sIgA preparations more efficiently hydrolyzed miR-200c-3p at two sites in this loop (Figure 4C). The most efficient hydrolysis of this miR-200c-3p by sIgAs and IgGs occurred at the 5A-6D site outlying from the loop (Figure 4C,D).

The double-stranded loop fragment of miR-378a-3p includes 16 of its 22 nucleotides (Figure 5A,B). In addition, 8 of 11 hydrolysis sites in this case are located in this specific loop. The relative average percentages of hydrolysis of miR-378a-3p by sIgAs (1.7–5.2%) and IgGs (1.6–4.4%) were relatively low. However, there were three hydrolysis sites of this microRNA in the region from 3G to 6G. In the case of several sIgA preparations, hydrolysis at the 4A-5A site proceeded more efficiently (12%).

Four of the eight hydrolysis sites of let-7f-5p are disposed in its loop, having no double-stranded regions (Figure 5C,D).

Of the six sites of pronounced hydrolysis, three are also located in a specific loop of miR-146b-5p. Some sIgA preparations hydrolyzed these microRNAs at two sites (7U-8G and 10U-11A) more efficiently than at other ones (Figure 5D). However, one sIgA preparation hydrolyzed this microRNA most efficiently at the 5U-6A site (24.4%). At the same time, the overall sites of maximum hydrolysis of let-7f-5p by sIgAs and IgGs were entirely different: 5U-6A and 7U-8G (Figure 5C,D).

sIgAs and IgGs hydrolyzed miR-146b-5p at six sites, three of which are located in the loop. Hydrolysis at these three loop sites was moderate (Figure 6).

The site of maximum hydrolysis of this microRNA by sIgAs and IgGs was 9C-10C6, adjacent to the loop (Figure 6). Unlike sIgAs (Figure 6A), several IgGs more efficiently hydrolyzed this microRNA at two sites: 5G-6A and 6A-7U (Figure 6B). Regardless of the absence or presence of double-stranded regions in specific loops, all five microRNAs were mainly hydrolyzed by sIgAs and IgGs at the sites of their particular loops. Hydrolyses at some sites of all five microRNAs in the case of sIgAs and IgGs were comparable; however, in the case of some sites, significant differences were observed. The most striking differences in the hydrolyses of miRNAs by sIgAs and IgGs were observed in the case of a specific hairpin fragment of miR-200c-3p (Figure 4C,D), 4A-5A site of miR-378a-3p (Figure 5A,B), 5U-6A site of let-7f-5p (Figure 5C,D), and 5G-6A and 6A-7U sites of miR-146b-5p.

## 3. Discussion

MicroRNAs regulate the expression of many genes through association with Argonaute [75,76,77,78,79,80,81]. At the same time, miRNAs are susceptible to degradation. Although association with Argonaute protects miRNAs from nucleases, extensive pairing with some unusual RNA targets can trigger miRNA hydrolysis [75,80]. It is believed that the degree of complementarity and the miRNA/target ratio are critical for efficient miRNA hydrolysis [75,77]. MiRNA and the target form a duplex with an unpaired flexible linker, which leads to duplex bending and opening of the 3′-end of miRNA for enzymatic attack [77]. According to [76], endogenous RNA (SERPINE1) controls the hydrolyses of two miRNAs (miR-30b-5p and miR-30c-5p) in mouse fibroblasts. Targeted miRNA degradation requires ZSWIM8 Cullin-RING E3 ubiquitin ligase [80]. Some studies have found other proteins involved in miRNA degradation, and it is also noteworthy that the unprotected 3′-ends of miRNAs can become available for enzymatic attack by 3′-5′-exonucleases, or by unidentified other cellular enzymes [75,78]. Thus, destruction occurs in a complex with Argonaute and miRNA at the 3′-end of the molecule.

It was previously shown that Abs from healthy volunteers could not split RNAs and DNAs [12,13,14,15,16,17]. However, IgGs and IgAs from human milk effectively hydrolyzed supercoiled DNAs and polymeric RNAs [12,13,14,15,16,17,54].

Cell-free microRNAs in different biological fluids have been revealed in membrane-coated microparticles (microvesicles, exosomes, ectosomes, etc.), as well as in membrane-free microparticles (high- and low-density lipoproteins, and were associated with different RNA-binding proteins and the surface of blood cells [82,83]. 

For the first time, the RNase activities of IgGs and sIgAs from a mother’s milk in the hydrolysis of four microRNAs were described in [73,74]. It showed earlier that all IgGs and sIgAs effectively hydrolyze four microRNAs: miR-9-5p, miR-219-2-3p, miR-137, and miR-219a-5p, which are found in high concentrations in the blood of patients with autoimmune diseases [73,74]. Elevated concentrations of these four microRNAs are more characteristic of patients with SLE and MS, and IgGs effectively hydrolyzed them from the blood of these patients [69,70,71,72]. 

In this study, a quantitative analysis of the relative contents of five microRNAs (miR-148a-3p, miR-200c-3p, miR-378a-3p, let-7f-5p, and miR-146b-5p), which are present in elevated concentrations in human milk, was carried out for the first time. Interestingly, the average content of the sum of all five microRNAs decreased in the following order (number of copies in 1 mg of sample): lipid fractions (1.46 × 10^10^) ≈ cells (1.25 × 10^10^) > plasma (0.18 × 10^10^) (Table 2).

The statistical differences in almost all microRNA contents in cells, plasma, and lipid fractions, were mainly significant (*p* = 0.001–0.04), except for some of them (*p* > 0.05). Positive CCs between the five miRNAs in most preparations of milk plasma (0.14 to 0.92), cells (0.04 to 0.99), and lipid fractions, varied over a wide range, with several exceptions when negative correlations were observed from −0.005 to −0.37 (Table 1).

It was interesting how the content of each specific microRNA correlated in three different milk fractions. It turned out that the content of miR-148a-3p (0.49–0.94), let-7f-5p (0.18–0.3), miR-146b-5p (0.12–0.62), and miR-378a-3p (0.25–0.92) in plasma, cells, and lipid fractions, were characterized mainly by positive CCs (Table 1). Three CCs characterizing the content of miR-200c-3p in three types of preparations turned out to be very different: 0.54 (cell–lipid fractions), 0.003 (plasma–lipid fractions), and −0.22 (plasma–cell fractions) (Table 1).

IgGs and sIgAs were isolated from milk plasma, and their relative catalytic activities in the hydrolysis of five microRNAs were estimated (Table 3). It should be noted that the relative activities of IgGs and sIgAs in the hydrolysis of five microRNAs depended very strongly on the milk preparation. For example, the rate of hydrolysis of miR-200c-3p by IgG7 was 21.3 times higher than that for IgG2, while sIgA2 hydrolyzed this microRNA 20.2 times slower than sIgA1. Overall, the efficiencies of some microRNAs hydrolyzed by seven IgGs and sIgAs did not correspond to the normal Gaussian distribution. CCs of the hydrolyses of five RNAs by the seven IgGs ranged from 0.17 to 0.82, while for seven sIgAs from −0.08 to 0.98 (Table 3). The CCs between concentrations of five microRNAs in plasma were also very different, and varied from 0.07 to 0.99 (Table 3). CCs between microRNA concentrations in individual plasmas and RAs of Abs corresponding to these plasmas in the hydrolyses of five microRNAs by IgGs varied from −0.01 to 0.79, and for sIgAs from 0.03 to 0.80 (Table 3). Milk was collected from seven women at about the same time after lactation began. However, in the case of three RNAs, the CCs between the RAs of IgGs and sIgAs were negative (−0.07–−0.41), and for two RNAs, they were highly positive (0.87) (Table 3).

The statistical differences in the content of almost all microRNAs in the cells were significant (*p* = 0.001–0.003), except for 148a-3p-278a-3p and 148a-3p-278a-3p (*p* = 0.79). A similar situation was found in the case of the lipid fractions (*p* = 0.001–0.04), except for two pairs of parameters: 148a-3p-278a-3p and 146b-5p-let-7f-5p (*p* = 0.1–0.96). The differences in the content of most microRNAs in milk plasma were also high (*p* = 0.001–0.01), except for 146b-5p-200c-3p and 148-3p-378-3p (*p* = 0.43–0.87).

The sites of five microRNAs hydrolyzed by sIgAs and IgGs are indicated on RNA’s spatial structures (Figure 4, Figure 5 and Figure 6). Substantially, sIgAs and IgGs hydrolyzed five microRNAs at the same sites, mainly located in the loop structures of the substrates (Figure 4, Figure 5 and Figure 6). The hydrolysis efficiencies of five RNAs by sIgAs and IgGs were predominantly comparable. At the same time, several of the seven sIgA and IgG preparations particularly hydrolyzed these microRNAs more efficiently than others. Hydrolysis of microRNAs occurs after their various bases: A, G, U, and C (Figure 4, Figure 5 and Figure 6). No pronounced specificity of microRNA hydrolysis after a specific base was observed. It is possible that, in principle, the spatial structures of different microRNAs are, to some extent, a more critical factor in determining cleavage sites.

As mentioned above, the destruction of microRNAs occurs in a complex with Argonaute at the 3′-end of RNAs. It should be assumed that some part of microRNAs in milk can exist in a free form. These microRNAs can be site-specific, and hydrolyzed by IgGs not at their 3-terminus.

As shown in previously published studies [74,75] as well as this article, milk abzymes can hydrolyze miRNAs not only at specific sites, but also non-specifically at different sites. Non-coding miRNAs have been identified in a wide range of bacteria (including pathogenic species); they play an important role in the regulation of many processes, including toxin gene expression [83]. It is proposed that these bacterial miRNAs may contribute to the regulation of prokaryotic-cell based production of toxins. Thus, it cannot be ruled out that antibodies against miRNAs of human milk in newborn organisms can hydrolyze miRNAs of bacteria, including pathogenic ones. 

## 4. Materials and Methods

### 4.1. Chemicals and Donors

Most chemicals of high quality used for this study were obtained from Sigma (St. Louis, MO, USA). Protein A-Sepharose, Superdex 200 HR 10/30 column, and Protein G-Sepharose were provided by GE Healthcare (GE Healthcare, New York, NY, USA). FastAP thermosensitive alkaline phosphatase and RNase A were obtained from Fisher Scientific (Pittsburgh, PA, USA). Fluorescein isothiocyanate (FITC) was acquired from Thermo Fisher (Thermo Fisher; Waltham, MA, USA; New York, NY, USA). FITC conjugates of oligonucleotides (ONs) were synthesized using the solid phase phosphoramidite method [84]. According to their analysis, all ribo-ONs were homogeneous according to reversed-phase chromatography and electrophoresis in 20% polyacrylamide gel.

RNA was isolated using reagents, buffers, and columns, using a special Lira kit (BiolabMix, Novosibirsk, Russia; https://biolabmix.ru/catalog/nabory-i-reagenty-nk/nabor-dlya-nk/reagent-lira-nabor-lira-dlya-vydeleniya-rnk-dnk-i-belkov/; accessed on 5 January 2017). RNA isolation was carried out according to the instructions of this company.

The milk sampling protocol was confirmed by the human ethics committee of Novosibirsk State Medical University (Novosibirsk, Russia; number 105-HIV; 07. 2010). The ethics committee supported this study in compliance with Helsinki ethics committee guidelines. All mothers provided written agreement to donate their milk for scientific studies. The mothers had no history of gastrointestinal, respiratory, autoimmune, rheumatologic, cardiovascular, or other system pathologies. All studies of possible diseases in women and the collection of milk preparations were carried out by doctors of the State Budgetary Health Institution of the Novosibirsk Region “Maternity Hospital No. 7”. Further, milk preparations were collected by nurses and issued to the employees of the Institute of Chemical Biology and Fundamental Medicine (IChBFM), according to a special agreement between Maternity Hospital No. 7 and IChBFM.

### 4.2. Purification and Analysis of RNAs

All milk preparations were collected by Maternity Hospital No. 7 nurses 10–15 days after the start of lactation. RNA preparations were isolated from different fractions of human milk: cell sediments, lipid fractions, and milk plasma. A combination of special Lira reagents and buffers and columns (BiolabMix) was used for isolation. A special solution containing 1 mL of Lira reagents was added to 100 μL of the sample and incubated for 10 min at room temperature, with constant stirring. Then, 200 μL of chloroform was added, and the mixture was incubated for 10 min at room temperature. The resulting mixture was centrifuged for 12 min at 10,000× *g* at 4 °C. The aqueous upper phase was used for the isolation of RNAs. To the RNA-containing aqueous phase, an equal volume of 96% ethanol was added, and this mixture was applied to the silicon membrane of a particular column (BiolabMix). After column centrifugation in a special tube for 30 s at 10,000× *g*, at 4 °C, the filtrate realized from the column was removed. Then, to 150 μL of special Lira concentrated buffer WB1, 350 μL of ethanol was added, and 500 μL of the mixture was applied to the column. After centrifugation of the column for 30 s at 10,000× *g*, at 4 °C, the filtrate was removed. Then, this operation was repeated once more. For complete removal of the buffer, the column was centrifuged again for 3 min at 10,000× *g*. In order to remove RNAs from the column, 60 µL of special buffer for RNA elution was applied to the column, which was incubated at room temperature for 5 min, then centrifuged for 1 min at 10,000× *g*, at 4 °C. In order to remove co-extracted DNA, the obtained RNA preparations were incubated for 20 min at 37 °C with DNase I; for inactivation of DNase I, the solution was heated to 70 °C for 5 min.

### 4.3. RNA Amplification

RNA amplification was performed according to a standard protocol, using special primers on a Bio-Rad CFX Connect device (Hercules, CA, USA; New York, NY, USA).

In the first stage, microRNA reverse transcription was performed using specific stem-loop primers and a reverse transcription kit that included M-MuLV-RH reverse transcriptase (BiolabMix). The reaction mixture contained 2 μL of RNA, 1 μL of stem-loop reverse transcription (RT)-primer, and 12 μL of H_2_O; it was heated to 65 °C for 5 min, and then cooled on ice for 2 min. The resulting mixture was centrifuged; 1 µL of M-Mul V and 4 µL of M-MuI V buffer were added. The mixture was incubated in a Bio-Rad CFX Connect cycler: 30 min at 16 °C, followed at 30 °C for 30 s for 60 cycles, at 42 °C for 30 s, at 50 °C for 1 s, and finally at 85 °C for 5 min. Then, the reverse transcription products were analyzed via real-time polymerase chain reaction using a fluorescent probe. The reaction mixture (20 μL) contained 10 μL of BioMaster HS-qPCR (2×) (BiolabMix), 5.6 μL of H_2_O, 1 μL of forward primer (10 μM), 1 μL of reverse primer (10 μM), 1.0 μL of fluorescent probe (2.5 µM), and 2 µL of reverse transcription product. The reaction mixture was incubated in a Bio-Rad CFX Connect system at 95 °C for 5 min, followed for 45 cycles at 95 °C for 5 s, at 60 °C for 10 s, and at 72 °C for 1 min. For reverse transcription of the mRNA of the reference genes, the oligo (dT) primer and the SYBR intercalating dye were used instead of the stem-loop primer.

Primers:

Forward: GTCATCCCTGAGCTGAACGG

Reverse: TTGAGGGCAATGCCAGCC

The first step that involved reverse transcription of microRNAs was performed using oligo(dT) primers and a reverse transcription kit containing reverse transcriptase M-MuLV–RH (BiolabMix). The reaction mixture containing 2 μL of RNA, 1 μL of oligo(dT)-primer, and 12 μL of H_2_O, was heated to 65 °C for 5 min, and then cooled on ice for 2 min. After mixture centrifugation, 1 µL of M-Mul V and 4 µL of M-MuI V buffer were added. The mixture was incubated in a Bio-Rad CFX Connect cycler for 30 min at 16 °C, and then for 60 cycles at 30 °C for 30 s, at 42 °C for 30 s, at 50 °C for 1 s, and finally at 85 °C for 5 min. The reverse transcription products were analyzed using real-time PCR with SYBR intercalating dye detection. The 20 μL reaction mixture contained 10 μL of HS-qPCR SYBR Blue (2×) (BiolabMix), 6 μL of H_2_O, 1 μL of forward primer (10 μM), 1 μL of reverse primer (10 μM), and 2 μL of reverse transcription product. The reaction mixture was incubated in a Bio-Rad CFX Connect cycler at 95 °C for 5 min, followed by 45 cycles at 95 °C for 5 s, at 60 °C for 10 s, and at 72 °C for 1 min.

The calibration curves were obtained on the basis of the amplification data of synthetic microRNA that was used in a concentration from 10^−3^ to 10^−7^ ng/µL. Using applicator software and calibration curves, the concentrations of each studied microRNA (ng/µL) in different human milk fractions were calculated, and recalculated to the number of microRNA copies in 1 mg of the investigated fractions of human milk plasma, cells, and lipid fractions under study.

### 4.4. Purification and Analysis of Antibodies

The milk of 7 healthy females residing in the Russia Novosibirsk region (120 mL at one time, 20–35 years old, respectively) was collected at 30–34 days after the onset of lactation. Milk samples were collected using the standard sterile appliances intended for the collection of excess mother’s milk. For 1–3 h after collection, all samples were cooled to 4 °C, and centrifuged for 20 min at 14 thousand rpm using an Eppendorf centrifuge; cells, lipid phases, and milk plasma were obtained. Immunoglobulins were purified from each milk sample, in a manner similar to [63,64,65,66,67,68]. There were no substantial variations found in any analyzed parameters of antibodies and abzymes (relative content of Abs and their catalytic activities), within the sampling period of 1–4 weeks after the beginning of lactation [73,74].

In order to obtain IgGs, the milk plasma was delivered into a column with Protein G-Sepharose equilibrated with buffer A (20 mM Tris-HCl (pH 7.5), 0.15 M NaCl) as described in [73,74]. The flow-through fraction containing sIgA antibodies was applied into a column with Protein A-Sepharose equilibrated with buffer A. All nonspecifically adsorbed proteins were eluted from Protein G-Sepharose and Protein A-Sepharose, first using buffer A up to zero optical density (A_280_), then with buffer A supplemented with 0.3 M NaCl and 1.0% Triton X-100, and again with buffer A. IgGs and sIgAs were specifically eluted from each of the sorbents with 0.1 M glycine-HCl (pH 2.6), immediately neutralized using 1.0 M Tris-HCl (pH 8.5), and then dialyzed against 20 mM Tris-HCl (pH 7.5) [73,74].

For additional purification, sIgA and IgG preparations (1.0–5.0 mg/mL, 0.3 mL) were incubated in 20 mM glycine-HCl buffer (pH 2.6) supplemented with 0.2 M NaCl at 26 °C for 20–30 min. The Abs were additionally purified using FPLC gel filtration on a Superdex 200 HR, according to procedures described in [63,64,65,66,67,68]. Collected fractions were immediately neutralized using Tris-HCl buffer (pH 9.0), and then dialyzed against 20 mM Tris-HCl (pH 7.5). For refolding of Abs after acidic treatment, their RNase activities were measured after 7–14 days from samples stored in this buffer at 4 °C.

### 4.5. Analysis of MicroRNAs Hydrolysis by Abs

Fluorescently labeled miR-148a-3p (5′-Flu-UCAGUGCACUACAGAACUUUGU-3′), let-7f-5p (5′-Flu-UGAGGUAGUAGAUUGUAUAGUU-3′), miR-146b-5p (5′-Flu-UGAGAACUGAAUUCCAUAGGCUG-3′), miR-200c-3p (5′-Flu-UAAUACUGCCGGGUAAUGAUGGA-3′, and miR-378a-3p (5′-Flu-ACUGGACUUGGAGUCAGAAGGC-3′), containing fluorescent residue (fluorescein, Flu) on their 5′-terminus, were used.

These microRNAs were chosen, since they are found in human milk in high concentrations [5,9].

The reaction mixture (15 μL) contained 50 mM Tris-HCl buffer (pH 7.5), 0.01 mg/mL of labeled microRNA, and 40 µg/mL IgG or sIgA antibodies similar to [73,74]. Final mixtures were incubated for 1 h at 37 °C. The reactions were stopped by the addition of a special denaturing buffer (15–20 μL) containing 8.0 M urea and 0.025% xylene cyanol. The products of all RNAs that were hydrolyzed were analyzed using 20% PAGE and denaturing conditions (0.1 M boric acid, 0.1 Tris, 8.0 M urea, and 0.02 M Na_2_EDTA, pH 8.3). Each gel was analyzed with a Typhoon FLA 9500 laser scanner (GE Healthcare, New York, NY, USA). In order to obtain markers of ON length, limited statistical hydrolyses of microRNAs with unspecific alkaline RNase (splitting RNAs with comparable efficiency at all internucleoside bonds) and specific RNase T1 were carried out. The products after alkaline hydrolysis contained cyclic 3′-monophosphate, which possesses lower electrophoretic mobility; they provide additional bands. Therefore, they were treated with FastAP thermally sensitive alkaline phosphatase. All of the gels were analyzed using a Typhoon FLA 9500 laser scanner (GE Healthcare, New York, NY, USA). The results were reported in terms of the mean ± standard deviation of at least three independent experiments.

### 4.6. Spatial Model of MicroRNAs

The spatial models of four microRNAs were generated by Predict a Secondary Structure server, which uses a combination of four algorithms for predicting the secondary structure of RNA, similarly to [69,70,71]: calculating a partition function, denoting the structure with minimal energy by http://rna.urmc.rochester.edu/RNAstructureWeb/Servers/Predict1/Predict1.html (accessed on 14 February 2018).

### 4.7. Statistical Analysis

The relative activities of the IgG and sIgA preparations were estimated from a decrease in the fluorescence intensity of the initial microRNA, in comparison with that of the control experiment which corresponded to incubation of five microRNAs without Abs. The results were reported as the mean and the standard deviation of at least two to three independent experiments for each IgG and sIgA preparation. Most of the sample sets did not meet the normal Gaussian distribution. In order to check for normality, Shapiro–Wilk’s W-test criterion was used. The correlation coefficients between different parameters were calculated using Shapiro–Wilk’s W-test criterion. The differences between different groups of microRNAs of various groups of IgG and sIgA sets were estimated by the Mann–Whitney test (Statistica 10; Statistical Package, StatSoft. Inc., Tulsa, OK, USA; http://www.statsoft.com/Products/STATISTICA-Features; StatSoft. Inc., New York, NY, USA, accessed on 20 January 2010); the value *p* < 0.05 was considered statistically significant. The median (M) and interquartile ranges (IQR) were estimated.

## 5. Conclusions

In summary, we quantified the relative concentrations of five microRNAs (miR-148a-3p, miR-200c-3p, miR-378a-3p, miR-146b-5p, and let-7f-5p) which are found in mother’s milk in high concentrations. It was shown that human milk polyclonal sIgAs and IgGs possess site-specific microRNA-hydrolyzing activities. It was shown that each milk preparation was characterized by a specific content of five microRNAs, which IgG and sIgA antibodies (abzymes) hydrolyzed. The correlation coefficients of the content of five microRNAs in three fractions of milk (cells, plasma, and lipid fractions) were determined, and the content of microRNAs in plasma with relative activities of sIgAs and IgGs in the hydrolyses of these five RNAs varied over a wide range, from negative to positive.

## Figures and Tables

**Figure 1 ijms-23-12070-f001:**
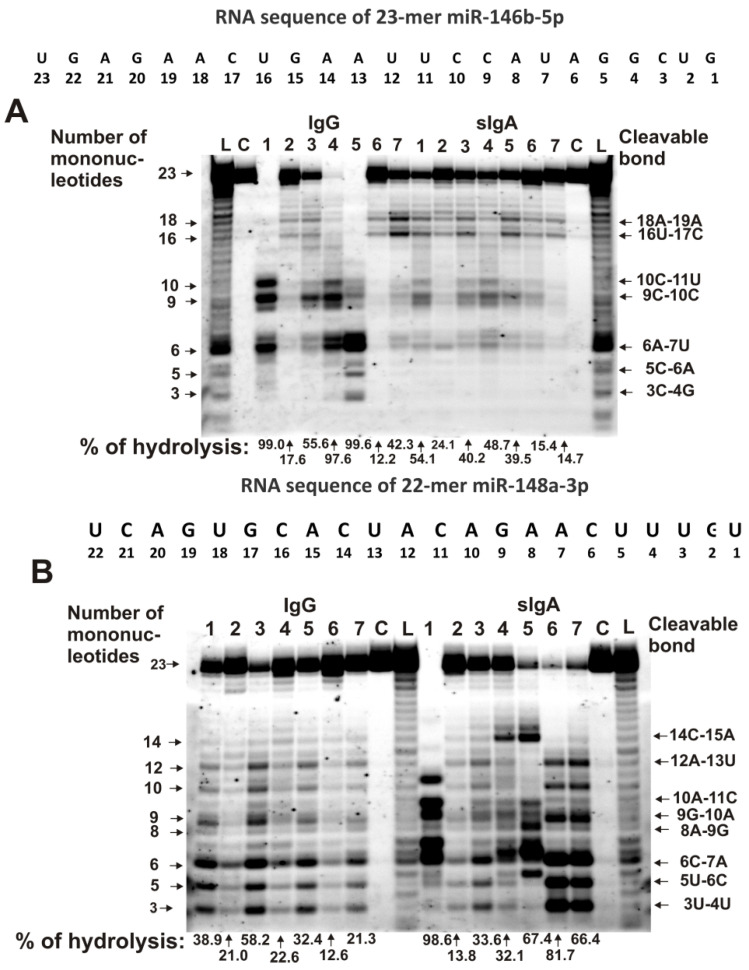
The patterns of 0.01 mg/mL Flu- miR-146b-5p (**A**) and Flu- miR-148a-3p (**B**) hydrolysis by sIgAs and IgGs (0.04 mg/mL) from human milk plasma. The hydrolysis products were detected due to the fluorescent residue (Flu) on microRNAs’ 5′-ends after reaction mixture incubation for 1 h. The numbers of sIgAs and IgGs, lengths of the splitting products, and the percentages of hydrolysis of two microRNAs by each antibody preparation, are indicated in panels (**A**,**B**).

**Figure 2 ijms-23-12070-f002:**
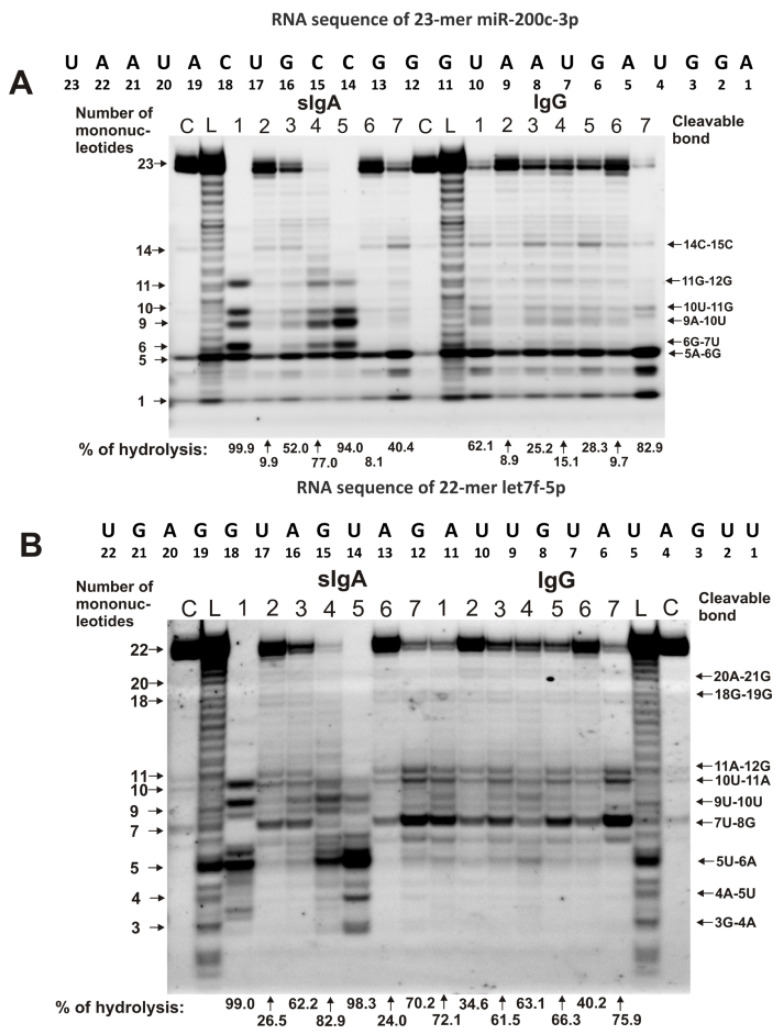
The patterns of 0.01 mg/mL Flu-miR-200c-3p (**A**) and Flu-let-7f-5p (**B**) hydrolysis by sIgAs and IgGs (0.04 mg/mL) from human milk plasma. The hydrolysis products were detected due to the fluorescent residue (Flu) on microRNAs’ 5′-ends after reaction mixture incubation for 1 h. The numbers of sIgAs and IgGs, lengths of the splitting products, and the percentages of hydrolysis of two microRNAs by each antibody preparation, are indicated in panels (**A**,**B**).

**Figure 3 ijms-23-12070-f003:**
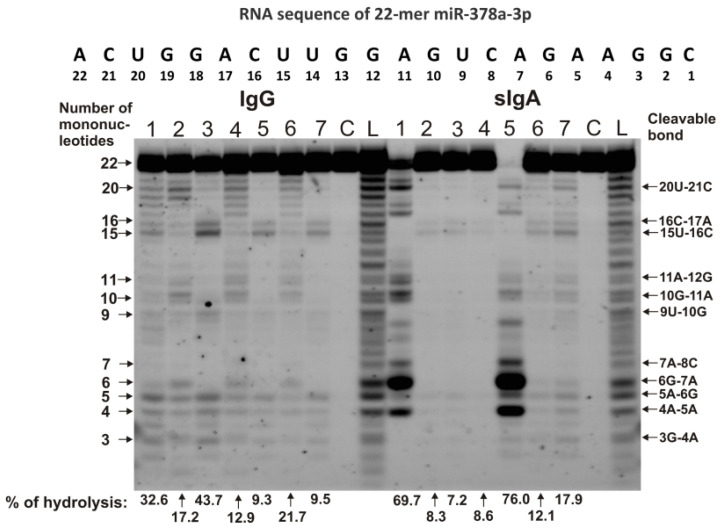
The patterns of 0.01 mg/mL Flu-miR-378a-3p hydrolysis by sIgAs and IgGs (0.04 mg/mL) from human milk plasma. The hydrolysis products were detected due to the fluorescent residue (Flu) on microRNAs’ 5′-ends after reaction mixture incubation for 1 h. The numbers of sIgAs and IgGs, lengths of the splitting products, and the percentages of hydrolysis of microRNA by each antibody preparation, are indicated in the panel.

**Figure 4 ijms-23-12070-f004:**
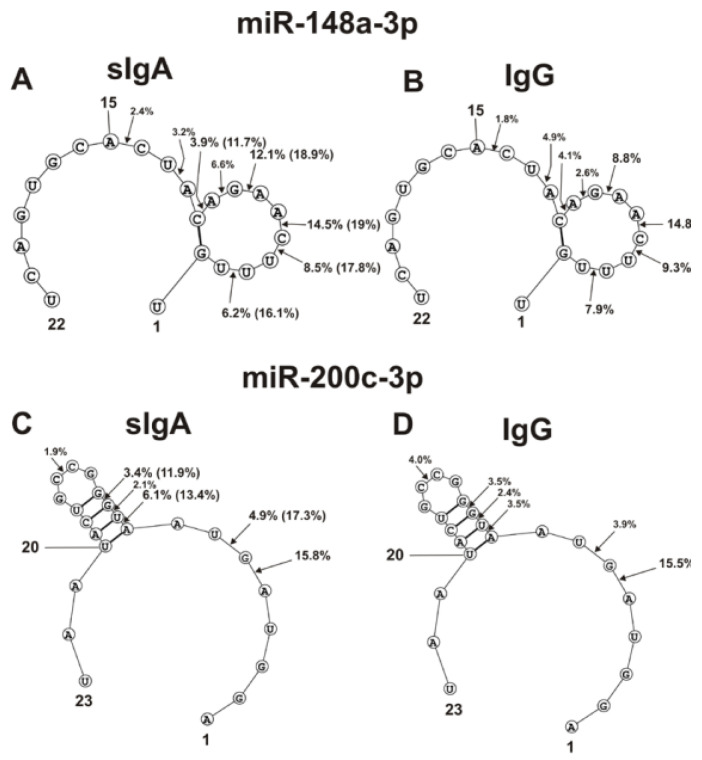
The average efficiencies of Flu-miR-148a-3p (**A**,**B**) and Flu-miR-200c-3p (**C**,**D)** hydrolyzed by seven sIgAs (**A**,**C**) and IgGs (**B**,**D**) from human milk plasma, in all sites of their cleavage. The average percentages of microRNAs and positions of separation in different sites of microRNA hydrolyzed by antibodies are shown using spatial models of microRNAs. Values in parentheses indicate the efficiencies of microRNA hydrolysis at individual sites by some preparations, with an increased specific activity in relation to these sites. Numbers 1, 15, 20, 22 and 23 correspond to the terminal RNA nucleotides.

**Figure 5 ijms-23-12070-f005:**
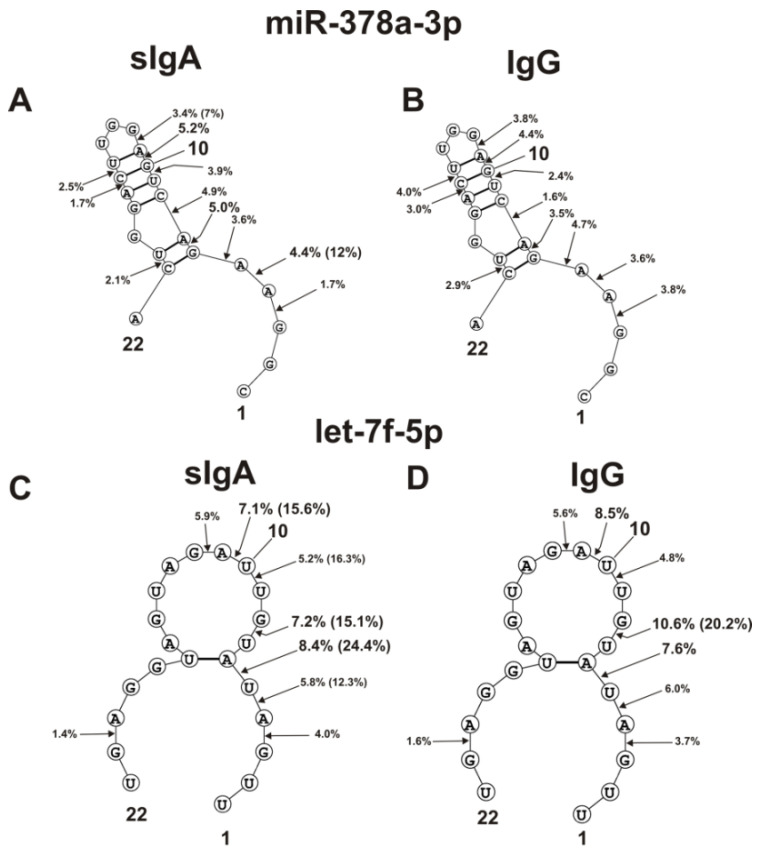
The average efficiencies of Flu-miR-378a-3p (**A**,**B**) and Flu-let-7f-5p (**C**,**D**) hydrolyzed by seven sIgAs (**A**,**C**) and IgGs (**B**,**D**) from human milk plasma, in all sites of their cleavage. The average percentages of microRNAs and cleavage positions in different microRNA hydrolysis sites by antibodies are shown using spatial models of microRNAs. Values in parentheses indicate the efficiencies of microRNA hydrolysis at individual sites by some preparations, with an increased specific activity in relation to these sites. Numbers 1, 10, 22 correspond to the terminal RNA nucleotides.

**Figure 6 ijms-23-12070-f006:**
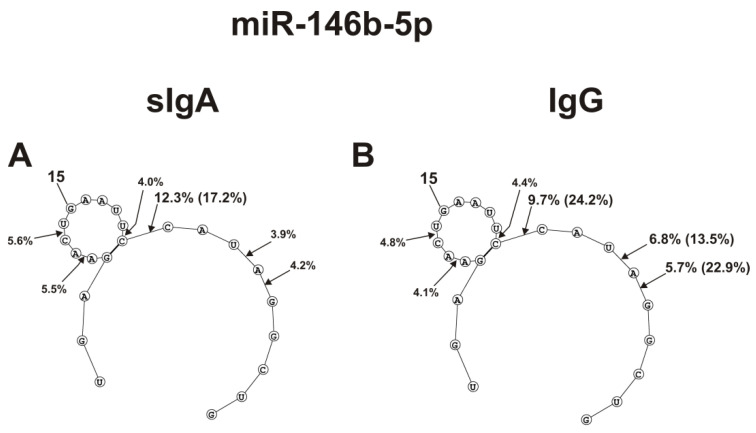
The average efficiencies of Flu-miR-146b-5p (**A**,**B**) hydrolyzed by seven sIgAs (**A**) and IgGs (**B**) from human milk plasma, in all sites of their cleavage. The average percentages of microRNAs and cleavage positions in different microRNA hydrolysis sites by antibodies are shown using spatial models of microRNAs. Values in parentheses indicate the efficiencies of microRNA hydrolysis at individual sites by some preparations, with an increased specific activity in relation to these sites. Number 15 corresponds to the nucleotide of the central part of the RNA loop.

**Table 1 ijms-23-12070-t001:** MicroRNA content in 1 mg of individual preparations of milk plasma, cells, and lipid fractions, and correlation coefficients (CCs).

Number of Donor	MicroRNA Content (Number of Copies in 1 mg of Milk Plasma)				
	miR-148a-3p	let-7f-5p	miR-146b-5p	miR-200c-3p	miR-378a-3p
	Number of Parameter				
	p1	p2	p3	p4	p5
1	1.2 × 10^8^ *****	1.4 × 10^10^	1.2 × 10^9^	3.7 × 10^9^	6.5 × 10^7^
2	8.0 × 10^6^	6.2 × 10^9^	1.1 × 10^9^	1.4 × 10^8^	2.8 × 10^7^
3	1.8 × 10^7^	5.2 × 10^9^	8.6 × 10^8^	1.5 × 10^9^	7.0 × 10^6^
4	6.0 × 10^6^	1.0 × 10^10^	1.1 × 10^9^	1.2 × 10^8^	3.0 × 10^6^
5	1.3 × 10^8^	4.8 × 10^9^	8.7 × 10^8^	4.2 × 10^9^	1.4 × 10^8^
6	1.1 × 10^8^	2.5 × 10^9^	4.2 × 10^8^	3.8 × 10^9^	9.8 × 10^7^
7	1.3 × 10^7^	4.9 × 10^9^	9.6 × 10^8^	5.7 × 10^7^	4.0 × 10^6^
CCs between parameters	1–2 (0.20); 1–3 (**−0.005**); 1–4 (0.67); 1–5 (0.92); 2–3 (0.81); 2–4 (**−0.19**); 2–5 (**−0.07**); 3–4 (0.58); 3–5 (0.14); 4–5 (0.66)				
Number of donor	MicroRNA content in cells (number of copies in 1 mg of milk cells)				
	miR-148a-3p	let-7f-5p	miR-146b-5p	miR-200c-3p	miR-378a-3p
	Number of parameter				
	p1	p2	p3	p4	p5
1	8.1 × 10^7^	5.0 × 10^10^	1.2 × 10^10^	9.2 × 10^9^	1.1 × 10^8^
2	1.2 × 10^8^	4.4 × 10^10^	9.4 × 10^9^	5.4 × 10^9^	2.0 × 10^8^
3	5.8 × 10^7^	4.4 × 10^10^	1.2 × 10^10^	2.6 × 10^9^	1.5 × 10^8^
4	9.7 × 10^8^	8.1 × 10^10^	2.3 × 10^10^	4.0 × 10^9^	1.2 × 10^8^
5	3.2 × 10^8^	5.5 × 10^10^	2.0 × 10^10^	5.3 × 10^9^	4.1 × 10^8^
6	8.8 × 10^8^	4.5 × 10^10^	1.1 × 10^10^	2.4 × 10^9^	1.6 × 10^8^
7	4.2 × 10^7^	3.7 × 10^10^	1.0 × 10^10^	2.2 × 10^9^	2.1 × 10^8^
CCs between parameters	1–2 (0.24); 1–3 (0.23); 1–4 (0.28); 1–5 (**−0.17**); 2–3 (0.99); 2–4 (0.86); 2–5 (**−0.37**); 3–4 (0.86); 3–5 (0.33); 4–5 (0.04)				
	MicroRNA content (number of copies in 1 mg of milk lipid fraction)				
Number of donor	miR-148a-3p	let-7f-5p	miR-146b-5p	miR-200c-3p	miR-378a-3p
	Number of parameter				
	p1	p2	p3	p4	p5
1	7.2 × 10^7^	4.1 × 10^10^	1.6 × 10^10^	3.1 × 10^9^	4.1 × 10^7^
2	1.3 × 10^7^	7.2 × 10^10^	1.6 × 10^10^	3.4 × 10^9^	3.7 × 10^7^
3	4.0 × 10^6^	2.4 × 10^10^	3.6 × 10^10^	1.6 × 10^9^	8.3 × 10^7^
4	6.2 × 10^8^	3.1 × 10^10^	6.4 × 10^9^	2.6 × 10^9^	7.1 × 10^7^
5	2.0 × 10^6^	1.0 × 10^10^	4.6 × 10^9^	5.9 × 10^8^	4.9 × 10^7^
6	4.1 × 10^8^	6.6 × 10^9^	2.0 × 10^9^	2.1 × 10^8^	2.3 × 10^7^
7	2.6 × 10^7^	7.4 × 10^9^	2.3 × 10^9^	1.9 × 10^9^	1.6 × 10^7^
CCs between parameters	1–2 (0.15); 1–3 (0.40); 1–4 (0.10); 1–5 (0.28); 2–3 (0.38); 2–4 (0.84); 2–5 (**−0.32**); 3–4 (0.29); 3–5 (0.26); 4–5 (0.48)				
CCs reflecting the content of each of the five RNAs (parameters p1–p5) in plasma, cells, and lipid fraction					
	p1 cells	p1 lipid fraction		p2 cells	p2 lipid fraction
1 plasma	0.57	0.49	2 plasma	0.18	0.30
1 cells	-	0.94	2 cells	-	0.26
	p3 cells	p3 lipid fraction	-	p4 cells	p4 lipid fraction
3 plasma	0.4	0.62	4 plasma	**−0.22**	0.003
3 cells	-	0.12	4 cells	-	0.54
	p5 cells	p5 lipid fraction	-	-	-
5 plasma	0.25	0.46	-	-	-
5 cells		0.92	-	-	-

***** For each value, a mean of three measurements is reported; the error of the determination of values did not exceed 7–10%.

**Table 2 ijms-23-12070-t002:** Values of relative content of five microRNAs in different fractions of human milk.

MicroRNAs	Number of Copies in 1 mg of the Analyzed Sample					
	Cells		Lipid Fraction		Plasma	
	Diapason *****	Average Value (M and IQR) ******	Diapason	Average Value (M and IQR)	Diapason	Average Value (M and IQR)
miR-148a-3p	4.2 × 10^7^–9.7 × 10^8^	3.8 × 10^8^ ± 1.3 × 10^8^ (2.2 × 10^8^; 5 × 10^8^)	2.0 × 10^6^–6.0 × 10^8^	2.0 × 10^8^ ± 9.0 × 10^7^ (5 × 10^7^; 4 × 10^8^)	2 × 10^6^–1 × 10^8^	5.1 × 10^7^ ± 2.0 × 10^7^ (1.6 × 10^7^; 1.0 × 10^8^)
miR-200c-3p	1.0 × 10^9^–9.0 × 10^9^	4.0 × 10^9^ ± 9.0 × 10^8^ (3.0 × 10^9^; 4.2 × 10^9^)	2.1 × 10^8^–3.4 × 10^9^	1.8 × 10^9^ ± 4.0 × 10^8^ (1.7 × 10^9^; 2.0 × 10^9^)	6.0 × 10^7^–4.0 × 10^9^	2.0 × 10^9^ ± 7.0 × 10^8^ (3.0 × 10^9^; 3.6 × 10^9^)
miR-378a-3p	6.2 × 10^7^–4.1 × 10^9^	1.8 × 10^8^ ± 3.7 × 10^7^ (1.5 × 10^8^; 5.0 × 10^8^)	2.0 × 10^7^–8.0 × 10^7^	4.0 × 10^7^ ± 8.0 × 10^6^ (4.0 × 10^7^; 4.0 × 10^8^)	2.0 × 10^6^–1.0 × 10^8^	4.0 × 10^7^ ± 2.0 × 10^7^ (2.0 × 10^7^; 1.0 × 10^8^)
let-7f-5p	1.8 × 10^10^–8.1 × 10^10^	4.6 × 10^10^ ± 6.0 × 10^9^ (4.5 × 10^10^; 3.2 × 10^10^)	4.1 × 10^9^–7.2 × 10^10^	2.5 × 10^10^ ± 8.0 × 10^9^ (1.7 × 10^10^; 2.4 × 10^10^)	2.1 × 10^9^–1.4 × 10^10^	6.2 × 10^9^ ± 1.0 × 10^9^ (5.0 × 10^9^; 3.6 × 10^9^)
miR-146b-5p	4.6 × 10^9^–2.3 × 10^10^	1.2 × 10^10^ ± 2.3 × 10^9^ (1.2 × 10^10^; 7.8 × 10^9^)	1.1 × 10^9^–3.6 × 10^10^	4.6 × 10^10^ ± 6.0 × 10^9^ (5.4 × 10^9^; 1.4 × 10^10^)	3.3 × 10^8^–1.2 × 10^9^	8.5 × 10^8^ ± 1.0 × 10^8^ (9.2 × 10^8^; 6.7 × 10^8^)
Average content of all microRNAs		1.25 × 10^10^ ± 1.93 × 10^10^		1.46 × 10^10^ ± 2.4 × 10^10^		0.18 × 10^10^ ± 0.04 × 10^10^

***** For each value, a mean of three measurements is reported; the error of the determination of values did not exceed 7–10%. ****** The median (M) and interquartile ranges (IQR) were calculated.

**Table 3 ijms-23-12070-t003:** Values of the relative activities of seven IgGs and sIgAs in the hydrolysis of five microRNAs, their average values, and correlation coefficients (CCs), and differences between groups of parameters (*p*).

Abs Number	miR-148a-3p	miR-200c-3p	miR-378a-3p	let-7f-5p	miR-146b-5p
	Number of Parameter				
	p1	p2	p3	p4	p5
IgG1	39.0 *****	62.1	69.8	72.1	99.5
IgG2	21.0	3.9	8.2	34.6	17.6
IgG3	58.2	25.2	7.2	61.5	55.6
IgG4	22.6	15.1	8.5	63.1	97.6
IgG5	32.4	28.3	76.1	66.3	99.7
IgG6	12.7	4.7	12.1	40.21	12.3
IgG7	21.3	82.9	17.9	76.0	42.3
Average values	29.6 ± 15.2	31.7 ± 29.9	28.5 ± 30.6	59.1 ± 15.7	60.7 ± 38.6
M; (IQR) ******	22.6 (18.0)	25.2 (57.4)	12.1 (61.6)	63.1 (31.9)	55.6 (81.9)
Correl. Coef.	1–2 (0.17); 1–3 (0.21); 1–4 (0.39); 1–5 (0.40); 2–3 (0.37); 2–4 (0.83); 2–5 (0.29); 3–4 (0.48); 3–5 (0.66); 4–5 (0.71)				
Difference (*p*)	1–2 (0.002); 1–3 (0.002); 1–4 (0.02); 1–5 (0.002); 2–3 (0.002); 2–4 (0.03); 2–5 (0.002); 3–4 (0.03); 3–5 (0.002); 4–5 (0.002)				
	Number of parameter				
Group number	6	7	8	9	10
sIgA1	98.7	99.0	32.6	99.0	54.1
sIgA2	13.8	4.9	17.2	26.5	24.0
sIgA3	33.7	52.0	43.7	62.2	40.2
sIgA4	32.1	77.1	12.9	83.0	48.7
sIgA5	67.4	94.0	9.2	98.3	39.5
sIgA6	81.8	4.1	21.7	24.0	15.3
sIgA7	66.4	40.4	9.4	70.2	14.7
Average values	56.3 ± 30.5	53.1 ± 39.2	21.0 ± 12.9	66.2 ± 31.0	33.8 ± 15.9
M and (IQR)	66.4 (98.7)	52.0 (99.0)	17.2 (43.7)	70.2 (99.0)	39.5 (54.1)
Correl. Coef.	6–7 (0.3); 6–8 (0.04); 6–9 (0.37); 6–10 (0.06); 7–8 (0.04); 7–9 (0.98); 7–10 (0.84); 8–9 (−0.08); 8–10 (0.36); 9–10 (0.73)				
Difference (*p*) *******	6–7 (0.7); 6–8 (0.002); 6–9 (0.7); 6–10 (0.03); 7–8 (0.002); 7–9 (0.4); 7–10 (0.03); 8–9 (0.7); 8–10 (0.03); 9–10 (0.03)				
CC between IgGs and sIgAs	1–6 (−0.07); 2–7 (−0.41); 3–8 (−0.09); 4–9 (0.87); 5–10 (0.87)				
Difference between IgGs and sIgAs (*p*)	1–6 (0.002); 2–7 (0.70); 3–8 (0.002); 4–9 (0.50); 5–10 (0.03)				

***** For each value, a mean of three measurements is reported; the error of the determination of values did not exceed 7–10%. ****** The median (M) and interquartile ranges (IQR) were calculated. ******* The differences between different groups of IgGs and sIgAs were estimated using the Mann–Whitney test.

**Table 4 ijms-23-12070-t004:** Values of correlation coefficients between concentrations of five microRNAs in human milk plasma *****.

MicroRNAs	miR-200c-3p	miR-378a-3p	let-7f-5p	miR-146b-5p
**miR-148a-3p**	0.97	0.99	0.07	0.33
**miR-378a-3p**	0.98	-	0.07	0.33
**let-7f-5p**	0.007	0.07	-	0.79
**miR-146b-5p**	0.42	0.33	0.79	-

***** The correlation coefficients between different parameters were calculated.

**Table 5 ijms-23-12070-t005:** Values of correlation coefficients between plasma microRNA concentrations, and relative activities of seven IgGs and sIgAs in the hydrolyses of five microRNAs *****.

Antibodies	MicroRNAs				
	miR-148a-3p	miR-200c-3p	miR-378a-3p	let-7f-5p	miR-146b-5p
IgGs	−0.01	−0.05	0.79	0.41	0.53
sIgAs	0.8	0.33	0.03	0.59	0.59

***** The correlation coefficients between different parameters were calculated.

## Data Availability

The data supporting our study results are included in the article.

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
