# Peer review of "MicroRNAs of Milk in Cells, Plasma, and Lipid Fractions of Human Milk, and Abzymes Catalyzing Their Hydrolysis"

_ijms, 2022, doi:10.3390/ijms232012070_

Round 1

Reviewer 1 Report

I have carefully with interest read the manuscript entitled: Maximally expressed miRNAs of milk in cells, plasma and lipid fraction of human milk and antibodies-abzymes catalyzing their hydrolysis.

The manuscript is interesting and well written. The methodology is described clearly. References are correctly presented.

Please send the consent of the bioethics committee and the consent of all study participants. In the text is information about milk samples but in Institutional Review Board Statement about blood samples: ,,The protocol of blood sampling was confirmed by the local Human ethics committee (Novosibirsk State Medical University, Novosibirsk, Russia; number 105-HIV; 07. 2010),,. This raises my doubts about the lack of proper ethical approvals for milk sampling collection and it should be verified. Please describe when the milk samples were collected (which day after delivery) and how mothers' health history without pathology was verified. If the authors had an agreement for the collection of medical information from patients/mothers?

Unfortunately, I have many doubts about ethics and committee approval for sample collection. Because of it, I can not recommend this paper for publication.

Below is one technical comment: Figure 1 has bad quality and is not readable.

Author Response

I have carefully with interest read the manuscript entitled: Maximally expressed miRNAs of milk in cells, plasma and lipid fraction of human milk and antibodies-abzymes catalyzing their hydrolysis.

The manuscript is interesting and well written. The methodology is described clearly. References are correctly presented.

Please send the consent of the bioethics committee and the consent of all study participants. In the text is information about milk samples but in Institutional Review Board Statement about blood samples: ,,The protocol of blood sampling was confirmed by the local Human ethics committee (Novosibirsk State Medical University, Novosibirsk, Russia; number 105-HIV; 07. 2010). This raises my doubts about the lack of proper ethical approvals for milk sampling collection and it should be verified. Please describe when the milk samples were collected (which day after delivery) and how mothers' health history without pathology was verified. If the authors had an agreement for the collection of medical information from patients/mothers?

Unfortunately, I have many doubts about ethics and committee approval for sample collection. Because of it, I can not recommend this paper for publication.

Below is one technical comment: Figure 1 has bad quality and is not readable.

Answer:

Figure 1 was corrected

Sorry, but according to the rules of the Russian Federation, we cannot send you the consent of the bioethics committee and the consent of all participants in the study. We, who conducted this study, were not doctors, but biochemists who received milk preparations from the State Budgetary Healthcare Institution of the Novosibirsk Region "Maternity Hospital No. 7" (GBUZ NSO "RD No. 7"), working together with the Medical University.

We have an agreement between our institute and Maternity Hospital No. 7 (which I am enclosing in Russian and translated into English in the “Unpublished results” section of the system for submitting an article to a journal. In addition, I cannot send you the consent of all participants in the study - these consents participants sign at Maternity Hospital No. 7. Doctors do not tell us the names of the mothers (this is prohibited by law), we receive milk products only with numbers - 1, 2, 3, etc. But according to the contract, the doctors undertake to conduct all studies of possible diseases of women in labor. I enclose a quote from the contract

“Hospital provides free of charge biological material, as well as the necessary consumables and other material for the collection and transportation of biological material: at least 10-20 blood products, milk and placenta from medically healthy women in labor (with a negative history of the following pathologies: HIV, hepatitis B and C, sexually transmitted infections, and other possible diseases)”.

As I noted above, I cannot send you the consent of all participants in the study - the participants sign these consents in Maternity Hospital No. 7 and their names are not disclosed. But I am sending you a consent form, which is signed by the women in labor in Russian and translated into English.

This work was carried out in full accordance with the rules of Helsinki ethics committee guidelines, however this information is the maximum that I can provide in accordance with Russian laws.

In “Materials and Methods” was added information on obtaining milk samples from "Maternity Hospital No. 7".

Sincerely,

Prof. Georgy Nevinsky.

Reviewer 2 Report

The authors of the manuscript "Maximally expressed miRNAs of milk in cells, plasma and lipid fraction of human milk and antibodies-abzymes catalyzing their hydrolysis" describe the concentration of specific miRNAs present in different fractions of human milk and how these miRNAs are hydrolyzed by milk IgG and IgA. While the scientific methodology appears sound, the manuscript needs considerable revision to make it more readable and less convoluted. Authors can benefit from manuscript editing help.

Here are a few suggestions:

1) Simplify the title. It is too wordy.

2) The abstract contains sentences that are very difficult to follow (for instance, lines 10-11). It should be revised.

3)Introduction is too long and covers topics that I don't believe are relevant to the research.

4) Please state the research objective and hypothesis at the end of the introduction

5) Methods: indicate the lactation stage of the milk collected from the women. miRNA concentration may vary with the lactation stage.

6) How do the forms of IgG and IgA differ?

7) Tables contain a lot of information. It may be beneficial to create more tables to describe the data.

8) The tables show parameters. Are the parameters the different miRNA? If so, why not just use the miRNA identification?

9) I assume the diapason refers to min and max values in Table 2. Median and IQR are also included. I would suggest choosing one of the other.

10) Correlation coefficients were calculated for miRNA, milk fraction, IgG, and IgA hydrolysis, many of which were very weak. One may question the reasons for so many correlations, which did not add much to understanding why miRNA are hydrolyzed.   

11) Discussion seemed to repeat the results. It would be interesting to suggest the impact of miRNA hydrolysis on the infant receiving mother's milk.

12) Discussion is very confusing as sometimes it refers to 4 miRNA studies when there were 5. See line 399. Table 4 should be reviewed.

Author Response

The authors of the manuscript "Maximally expressed miRNAs of milk in cells, plasma and lipid fraction of human milk and antibodies-abzymes catalyzing their hydrolysis" describe the concentration of specific miRNAs present in different fractions of human milk and how these miRNAs are hydrolyzed by milk IgG and IgA. While the scientific methodology appears sound, the manuscript needs considerable revision to make it more readable and less convoluted. Authors can benefit from manuscript editing help.

Here are a few suggestions:

  • Simplify the title. It is too wordy.

Answer: it was done

MicroRNAs of Milk in Cells, Plasma and Lipid Fraction of Human Milk and Abzymes Catalyzing Their Hydrolysis

  • The abstract contains sentences that are very difficult to follow (for instance, lines 10-11). It should be revised.

Answer: It was done

  • Introduction is too long and covers topics that I don't believe are relevant to the research.

Answer:

Sorry, but the description of abzymes in the literature is not frequent. Since we are dealing with different abzymes, we constantly receive additional questions from reviewers on the general situation in the study of abzymes. In addition, the literature mainly describes the abzymes of patients with autoimmune diseases. Some of the abzymes in human milk are very different from abzymes in patients with autoimmune diseases. With this in mind, in the introduction we try to describe what we usually add in accordance with the frequent comments of the reviewers of our articles.

  • Please state the research objective and hypothesis at the end of the introduction

Answer: it was done

  • Methods: indicate the lactation stage of the milk collected from the women. miRNA concentration may vary with the lactation stage.

Answer:

It was done: The milk of 7 healthy females residing in the Russia Novosibirsk region (120 mL at one time; 20–35 years old) was collected at 30-34 days after the onset of the lactation.

  • How do the forms of IgG and IgA differ?

Answer: Methods of purification and characterization, including electrophoretic homogeneity of IgGs (150 kDa, H2L2: two light (L) and two heavy (H) chains) and sIgAs (340 kDa,  (H2L2)2SJ: four light and four heavy chains, secretory (S) and join (J) components) preparations used in this study, were described in [73,74].

  • Tables contain a lot of information. It may be beneficial to create more tables to describe the data.

Answer: Sorry, but if you make more Tables, it will be very difficult to compare the difference in the content of components in cells, plasma and lipid fraction.

  • The tables show parameters. Are the parameters the different miRNA? If so, why not just use the miRNA identification?

Answer:

Sorry, I didn't understand the question. Each Table lists the parameters that characterize all five miRNAs indicated in the tables.

  • I assume the diapason refers to min and max values in Table 2. Median and IQR are also included. I would suggest choosing one of the other.

Answer:

Sorry according to our publication experience, if you include only one of the parameters, there are usually reviewers who ask us to provide both of the parameter. For different scientists, first parameter is more important, and for others, the secondo one.

  • Correlation coefficients were calculated for miRNA, milk fraction, IgG, and IgA hydrolysis, many of which were very weak. One may question the reasons for so many correlations, which did not add much to understanding why miRNA are hydrolyzed. 

Answer:

Sorry, but this is an analysis that needs to be carried out and such data are obtained, it is not possible to change them. Changes in the immune system of each woman during pregnancy and lactation are individual and occur in many different directions. As a result of the combination of various factors, we can get what we see from the experiment. This is what is called a real given. 

  • Discussion seemed to repeat the results. It would be interesting to suggest the impact of miRNA hydrolysis on the infant receiving mother's milk.

Answer:

As we have shown earlier, the catalytic activity of some abzymes in human milk can be easily explained. For example, abzymes hydrolyzing nucleic acids, peptides and proteins of viruses and harmful bacteria protect children due to passive immunity in the first period up to 6 months, while the child's immune system is not yet formed. From our point of view, it is still premature to discuss the question of the significance of abzymes hydrolyzing miRNAs. However we can propose possible role of abzymes of human milk.

Additional information:

As shown in previously published [74,75] and this article, milk abzymes can hydrolyze miRNAs not only at specific sites, but also non-specifically at different sites. Non-coding miRNAs have been identified in the wide range of bacteria (including pathogenic species); they play an important role in the regulation of many processes, including toxin gene expression [83]. It is proposed that these bacterial miRNAs may contribute in the regulation of prokaryotic-cell based production of toxins. Thus, it cannot be ruled out that antibodies against miRNAs of human milk in newborn organisms can hydrolyze miRNAs of bacteria, including pathogenic ones.   

  • Discussion is very confusing as sometimes it refers to 4 miRNA studies when there were 5. See line 399. Table 4 should be reviewed.
  • Answer:

Sorry, 4 miRNAs were analyzed previously and it was corrected

Sorry, but Table 4 has all five miRNAs. The first microRNA (miR-148a-3p ) is in the first column, and the first row shows its correlation coefficients with four other microRNAs. Such a table occupies the minimum volume.

Sincerely,

Prof. Georgy Nevinsky

Reviewer 3 Report

The manuscript focusses on milk components with an important biological function that may influence newborns' growth and development.  The current research study the interesting relationship between two types of milk components:  Ig and miRNAs. The results of this study are very interesting and can be the base of others research on the field.

Nevertheless, there is a need to discuss the biological function of the miRNA hydrolysis by Ig, and what may be the implication on newborns.

The authors must consider that miRNA in the milk "plasma" are protected and carried by extracellular vesicles such as exosomes. This point must be discussed in particular related to the explanation that Igs hydrolyzed miRNA protected by Argonaute.

The hydrolytic effect of milk Igs was demonstrated on "unprotected" miRNA. What is the expected effect of Igs on extracellular carried milk miRNAs? 

The authors must consider and discuss that the hydrolytic effect of milk Igs can be also on miRNA in the target cells of the newborn.

Fig 1 must be improved the quality of the image, impossible to read the numbers.

Line 450-453 why are highlighted?

Author Response

Comments and Suggestions for The manuscript focusses on milk components with an important biological function that may influence newborns' growth and development.  The current research study the interesting relationship between two types of milk components:  Ig and miRNAs. The results of this study are very interesting and can be the base of others research on the field.

  1. Nevertheless, there is a need to discuss the biological function of the miRNA hydrolysis by Ig, and what may be the implication on newborns.

Answer:

As we have shown earlier, the catalytic activity of some abzymes in human milk can be easily explained. For example, abzymes hydrolyzing nucleic acids, peptides and proteins of viruses and harmful bacteria protect children due to passive immunity in the first period up to 6 months, while the child's immune system is not yet formed. From our point of view, it is still premature to discuss the question of the significance of abzymes hydrolyzing miRNAs. However we can propose possible role of abzymes of human milk.

Additional information:

As shown in previously published [74,75] and this article, milk abzymes can hydrolyze miRNAs not only at specific sites, but also non-specifically at different sites. Non-coding miRNAs have been identified in the wide range of bacteria (including pathogenic species); they play an important role in the regulation of many processes, including toxin gene expression [83]. It is proposed that these bacterial miRNAs may contribute in the regulation of prokaryotic-cell based production of toxins. Thus, it cannot be ruled out that antibodies against miRNAs of human milk in newborn organisms can hydrolyze miRNAs of bacteria, including pathogenic ones.  

  1. The authors must consider that miRNA in the milk "plasma" are protected and carried by extracellular vesicles such as exosomes. This point must be discussed in particular related to the explanation that Igs hydrolyzed miRNA protected by Argonaute. The hydrolytic effect of milk Igs was demonstrated on "unprotected" miRNA. What is the expected effect of Igs on extracellular carried milk miRNAs? 

Answer: We  have included additional information.

Cell free microRNAs in different biological fluids revealed in membrane-coated microparticles (microvesicles, exosomes, ectosomes etc.) as well as in membrane-free microparticles (high- and low-density lipoproteins, and associated with different RNA-binding proteins and the surface of blood cells [82]. 

It seems premature to us to draw conclusions about how the hydrolysis of miRNA associated with proteins or within membrane-coated, membrane-free microparticles and low-density lipoproteins can occur. Complexes of proteins with RNA are in dynamic equilibrium with free RNA. In addition, as we have shown on the example of DNA associated with histones, it can be subjected to partial hydrolysis. with abzymes. Therefore, at the moment it is premature to evaluate how antibodies hydrolyzing free RNA or in some complexes work. Such conclusions may possibly be made in the future after a more detailed study of the availability of miRNAs in the composition of exosomes and complexes with proteins, which is a separate study.

The authors must consider and discuss that the hydrolytic effect of milk Igs can be also on miRNA in the target cells of the newborn.

Answer: As mentioned above, it cannot be ruled out that antibodies against miRNAs of human milk in newborn organisms can hydrolyze miRNAs of bacteria, including pathogenic ones.  

Fig 1 must be improved the quality of the image, impossible to read the numbers.

Answer: It was done

Sincerely,

Prof. Georgy Nevinsky

Round 2

Reviewer 1 Report

Dear Authors,
thank you for your explanation, however, your answer is not enough for me.

Russia does not respect Human Rights ( see reports of  Amnesty International) so I have many doubts about your declaration that work was carried out in full accordance with the rules of Helsinki ethics committee guidelines. Russia has been also suspended from Human Rights Council. The answer: ,,this information is the maximum that I can provide in accordance with Russian laws,, is not convincing. There must be transparency of procedures in scientific research and without doubt respect for human and patient rights. It is also an absolute requirement to publish the study results.

How long do healthy women stay in Maternity Hospital in Russia?
,,All milk preparations were collected by Maternity Hospital nurses 10-15 days after the start of lactation.,, - it is not clear. The first stage of lactation (lactogenesis) begins around the 16th week of pregnancy and lasts until a few days after you give birth, so the information is not precise. We know that the composition of the milk changes from day to day and depends on the health of the mother and the newborn baby. Who assessed it and how, under what conditions? Did healthy mothers with newborns stay in the hospital even 2 weeks after giving birth to collect milk samples for the study?

Author Response

Russia does not respect Human Rights ( see reports of  Amnesty International) so I have many doubts about your declaration that work was carried out in full accordance with the rules of Helsinki ethics committee guidelines. Russia has been also suspended from Human Rights Council. The answer: ,,this information is the maximum that I can provide in accordance with Russian laws,, is not convincing. There must be transparency of procedures in scientific research and without doubt respect for human and patient rights. It is also an absolute requirement to publish the study results.

Answer:

Sorry, but these are political, not scientific claims. It seems to me that it is not possible to prove to you that in Russia all scientists work in accordance with the Helsinki rules.

I sent you a form of agreement for parturient women to donate milk voluntarily for scientific purposes. You demand the original documents of the mothers in labor. But the consent documents contain the names of women in labor and their disclosure is prohibited both in Russia, and in America and Europe. Thus, I can not provide you with the original documents

Best wishes

Prof. Georgy A. Nevinsky

Reviewer 2 Report

I strongly recommend the use of academic journal editing.  The science is interesting, but the reading of the manuscript is still difficult and convoluted. 

Author Response

I strongly recommend the use of academic journal editing.  The science is interesting, but the reading of the manuscript is still difficult and convoluted. 

Answer: It was done

Sincerely

Best wishes

Prof. Georgy A. Nevinsky